# The Sink-Source Relationship in Cucumber (*Cucumis sativus* L.) Is Modulated by DNA Methylation

**DOI:** 10.3390/plants13010103

**Published:** 2023-12-28

**Authors:** Yudan Wang, Huimin Zhang, Jiawen Gu, Chen Chen, Jiexia Liu, Zhiping Zhang, Bing Hua, Minmin Miao

**Affiliations:** 1College of Horticulture and Landscape Architecture, Yangzhou University, Yangzhou 225009, China; dx120210135@stu.yzu.edu.cn (Y.W.); mz120211347@stu.yzu.edu.cn (J.G.); kukujujubo222@163.com (C.C.); 008302@yzu.edu.cn (J.L.); zhangzp@yzu.edu.cn (Z.Z.); binghua@yzu.edu.cn (B.H.); 2Jiangsu Yanjiang Institute of Agricultural Sciences, Nantong 226541, China; 20220057@jaas.ac.cn; 3Joint International Research Laboratory of Agriculture and Agri-Product Safety of Ministry of Education of China, Yangzhou University, Yangzhou 225009, China; 4Key Laboratory of Plant Functional Genomics of the Ministry of Education, Jiangsu Key Laboratory of Crop Genomics and Molecular Breeding, Yangzhou University, Yangzhou 225009, China

**Keywords:** 5-aza-dC-2′-deoxycytidine, cucumber, DNA methylation, raffinose family oligosaccharides, sink-source regulation

## Abstract

The optimization of the sink-source relationship is of great importance for crop yield regulation. Cucumber is a typical raffinose family oligosaccharide (RFO)-transporting crop. DNA methylation is a common epigenetic modification in plants, but its role in sink-source regulation has not been demonstrated in RFO-translocating species. Here, whole-genome bisulfite sequencing (WGBS-seq) was conducted to compare the nonfruiting-node leaves (NFNLs) and leaves of fruit setting (FNLs) at the 12th node by removing all female flowers in other nodes of the two treatments. We found considerable differentially methylated genes enriched in photosynthesis and carbohydrate metabolic processes. Comparative transcriptome analysis between FNLs and NFNLs indicated that many differentially expressed genes (DEGs) with differentially methylated regions were involved in auxin, ethylene and brassinolide metabolism; sucrose metabolism; and RFO synthesis pathways related to sink-source regulation. Moreover, DNA methylation levels of six sink-source-related genes in the pathways mentioned above decreased in leaves after 5-aza-dC-2′-deoxycytidine (5-Aza-dC, a DNA methyltransferase inhibitor) treatment on FNLs, and stachyose synthase (*CsSTS*) gene expression, enzyme activity and stachyose content in RFO synthesis pathway were upregulated, thereby increasing fruit length and dry weight. Taken together, our findings proposed an up-to-date inference for the potential role of DNA methylation in the sink-source relationship, which will provide important references for further exploring the molecular mechanism of DNA methylation in improving the yield of RFO transport plants.

## 1. Introduction

Crop yield depends on the potential of the sink (to store assimilates) and the capacity of the source (to output photosynthetic products) [1,2,3]. Mature leaves with net photosynthetic output capacity and fast-growing fruits that store assimilates are the most typical “source” and “sink” organs in fruiting crops. An inflated source/sink ratio may result in a waste of carbon assimilation, and a slow output of photosynthetic products from leaves will feed back to inhibit source assimilation activity [4,5]. On the other hand, a low source/sink ratio may lead to the consumption of assimilate from source organs and a decline in the photosynthetic area and yield [6]. Therefore, a reasonable source/sink ratio is crucial for achieving high crop yields. Compared to one-time harvest crops, the source/sink ratio of successive fruiting and harvested vegetables shows cyclical fluctuations and makes the sink-source balance more interesting and complex [7,8].

Carbon fixation rates and sucrose production in leaves have long been considered targets to improve crop productivity [9]. Sucrose is the primary form of carbohydrate transported from source organs (such as mature leaves) to sink organizations (such as roots, fruits, stems and seeds) via the veins in phloem tissues, and it also acts as an essential signaling molecule in plants [10]. In these sucrose-translocating crops, a higher source-sink ratio results in a decreased expression of genes and enzyme activity related to starch biosynthesis (ADP–glucose pyrophosphorylase, starch synthase, etc.) and sucrose catabolism (sucrose synthase, invertase, etc.) in leaves, while a lower source-sink ratio downregulates the expression of sucrose transporters (SUTs or SWEETs) in leaves, which are the key regulators of carbon partitioning between sources and sinks [11,12,13].

Cucumber (*Cucumis sativus* L.), a widely planted vegetable around the world, is one of the seven major vegetables with enormous economic value in China. The leaves and fast-growing fruits belong to the “source” and “sink” organs in cucumber. Cucumber is a typical raffinose family oligosaccharide (RFO)-transporting plant, determining that the sink-source regulation of cucumber differs from sucrose-transporting model plants, such as *Arabidopsis* (*Arabidopsis thaliana*), rice (*Oryza sativa*) and tomato (*Solanum lycopersicum*) [14,15,16,17]. The proportion of RFO in cucumber leaf assimilates is up to 80%, which is unique to RFO-transporting plants [18]. Photosynthetic products of RFO-translocating plants require an additional synthesis step from sucrose to raffinose and stachyose before being exported to sink organs [19,20,21]. Galactitol synthase (CsGolS1), raffinose synthase (CsRS) and stachyose synthase (CsSTS) are the main genes responsible for RFO biosynthesis in the phloem of cucumber leaves, perhaps also affecting the sink-source allocation [22,23]. A higher output rate of carbon is commonly followed by higher stachyose synthase (STS: EC.2.4.1.67) activity in leaves [24]. In recent years, some key genes that regulate the sink-source interaction at the transcriptional level have been identified. For example, Liu et al. [20] reported that the expression of alkaline α-galactosidase 2 (*CsAGA2*) in fruit is essential for cucumber sink strength and showed feedback regulation on source leaves. Dai et al. [16] demonstrated that the expression of *CsSTS* was significantly elevated in leaves with fast-growing fruit, promoting assimilate loading into the vascular tissue.

In addition to the transcriptional level, epigenetic modifications, such as DNA methylation, also play an indispensable role in regulating plant growth, development and yield formation. It has been widely reported that DNA methylation modulates gene expression in the pre- and post-transcriptional regulation of several biological processes, including the vernalization response, defense against parasitic elements, and stress responses [25,26,27,28,29]. Recent research has shown that DNA methylation affects the expression of genes related to starch and sucrose metabolism, photosynthesis, and plant hormone signal transduction, as well as stress response in sucrose-transporting crops, such as apple and sugarcane [30,31]. However, the role of DNA methylation in sink-source regulation in crops that rely on RFOs as the major transport form of photoassimilates in vivo has not been reported.

In order to explore the potential mechanism by which DNA methylation improves yield by modulating sink-source relationships, we determined the epigenetic genomic landscape of cucumber leaves with different sink strengths based on whole-genome bisulfite sequencing. We estimated the relationship between the enrichment level of DNA methylation and the transcription level of genes in auxin, ethylene and brassinolide biosynthesis and metabolism, as well as sucrose and RFO synthesis metabolism involved in the sink-source relationship. In addition, the leaves of fruit setting (FNLs) were treated with a DNA methylation inhibitor, 5-aza-dC-2′-deoxycytidine (5-Aza-dC). The leaves’ DNA methylation level of *CsRS* (*CsaV3_3G042690*), *AUX1* (*CsaV3_5G002130*), *SAUR* (*CsaV3_6G007970*), *ERF1* (*CsaV3_3G012170*), *BRI1* (*CsaV3_5G015150*) and *SUS* (*CsaV3_4G000970*); expression level of *CsGolS1* (*CsaV3_6G000050*), *CsRS* (*CsaV3_3G042690*) and *CsSTS* (*CsaV3_7G030090*); enzyme activity of CsGolS, CsRS and CsSTS; and soluble sugar content of leaves, as well as fruit size and dry weight, were also analyzed after 5-Aza-dC treatment on FNLs to understand the role of DNA methylation in cucumber sink-source relationships. This study provides new insights from an epigenetic point of view into the factors that affect sink-source relationships in Cucurbitaceae plants.

## 2. Results

### 2.1. Dynamic DNA Methylation Landscape of Whole-Genome Bisulfite Sequencing (WGBS) in the Nonfruiting-Node Leaves (NFNLs) and FNLs

As shown in Figure 1A, the sink strength of leaves in the FNL group is significantly higher than NFNLs. In the WGBS results, the average genome-wide methylation levels of CG, CHG and CHH sites in cucumber were 68.68%, 40.93% and 18.84% in NFNLs and 69.24%, 41.79% and 19.61% in FNLs, respectively (Figure 1B). Among the upstream 2 kb, the gene body, coding DNA sequence (cds), exon, and downstream 2 kb, upstream 2 kb showed the highest methylation levels in CG/CHG/CHH sites, the exon region showed the lowest CG methylation levels, and the cds region occupied the lowest proportion in CHG and CHH sites (Figure 1E). As shown in Figure 1C, 40, 7, and 6484 differentially methylated regions (DMRs) were identified between FNLs and NFNLs at the CG, CHG and CHH sites, respectively. The FNLs had 14 CG-DMRs, 2 CHG-DMRs and 3712 CHH-DMRs of upregulated genes and 26 CG-DMRs, 5 CHG-DMRs and 2772 CHH-DMRs of downregulated genes compared with the NFNLs (Appendix A). As shown in Appendix A, CHG-DMRs and CHH-DMRs were evenly distributed across the chromosomes, and CG-DMRs appeared on chr5, chr6 and chr7. The numbers of DMRs on different gene elements in CG/CHG/CHH sites are shown in Figure 1D. We found that intergenic regions existed in the majority of DMRs, followed by the upstream 2 kb, downstream 2 kb, the gene body, exons, and cds of the cucumber genome. Most DMRs were enriched in the CHH context.

### 2.2. Gene Ontology (GO) Analysis of Differentially Methylated Genes (DMGs)

The potential function of differentially methylated genes (DMGs) involved in sink-source carbon partitioning was investigated through gene ontology (GO) analysis. The results showed that the CG-DMGs in the FNLs vs. NFNLs comparison were significantly (*p* < 0.01) enriched in photosystem I (GO: 0009522) and photosynthesis (GO: 0015979) (Figure 2A), such as *PSI* (Photosystem I P700, *CsaV3_UNG203670* (down) and *CsaV3_UNG201530* (down)). Moreover, 98 genes were enriched in the carbohydrate metabolic process (*p* < 0.01) in CHH sites, such as *BGLU* (Glucan endo-1,3-beta-glucosidase; *CsaV3_5G000260* (up), *CsaV3_3G045050* (down), *CsaV3_1G014850* (up), *CsaV3_2G014660* (up), *CsaV3_1G015360* (down), etc.) and *GLU* (endoglucanase; *CsaV3_3G045760* (down), *CsaV3_6G037680* (down)), indicating that the methylation modification of these hydrolases may promote the decomposition of polysaccharides stored in FNLs to supply the growth of sink organs (Figure 2B).

### 2.3. Correlation between DNA Methylation and Gene Expression between FNLs and NFNLs

To analyze whether the changes in the methylation levels induced by different sink strengths were associated with gene expression, we analyzed transcriptome data generated from the same samples as those used for bisulfite-seq. To confirm the RNA-seq results, 11 genes, including auxin influx carrier (*AUX1*, *CsaV3_5G002130*), auxin-responsive protein (*IAA*, *CsaV3_2G013230*), small auxin up-regulated RNA10 (*SAUR*, *CsaV3_6G007970*), ethylene-responsive transcription factor 1 (*ERF1*, *CsaV3_3G012170*), brassinosteroid insensitive 1 (*BRI1*, *CsaV3_5G015150*), brassinosteroid insensitive 1-associated receptor kinase 1 (*BAK1*, *CsaV3_4G007620*), sucrose synthase 2-like (*SUS*, *CsaV3_4G000970*), sucrose-phosphate synthase 4 (*SPS*, *CsaV3_2G033300*), *CsGolS1* (*CsaV3_6G000050*), *CsRS* (*CsaV3_3G042690*), and *CsSTS* (*CsaV3_7G030090*), related to sink-source regulation, were selected for reverse transcription quantitative PCR (RT-qPCR) detection. Expression trends between the RT-qPCR results and the RNA-seq data agreed well (Figure 3 and Figure 4).

Among 1098 differentially expressed genes (DEGs) between FNLs and NFNLs, approximately one-third showed significant methylation level changes in CHH sites. Among the hypermethylated genes, 179 and 98 of the genes were upregulated and downregulated in FNLs, while 156 and 68 of the hypomethylated genes were upregulated and downregulated in FNLs, respectively. However, in CG sites, only four DEGs showed methylation differences between FNLs and NFNLs (Appendix A). Interestingly, we found that DMR-DEGs caused by sink-strength difference were enriched in auxin, ethylene and brassinolide biosynthesis and metabolism; sucrose metabolism; and RFO synthesis, including *AUX1* (*CsaV3_5G002130*), *IAA* (*CsaV3_2G013230*), *BRI1* (*CsaV3_5G015150*, *CsaV3_2G002530*), *ERF1* (*CsaV3_3G012170*), *SUS* (*CsaV3_4G000970*, *CsaV3_5G020420*), and *CsRS* (*CsaV3_3G042690*). The details of the DNA methylation and expression of these genes are shown in Figure 3 and Appendix A. Significantly, in the RFO synthesis pathway, *CsRS* was hypermethylated and upregulated in FNLs, indicating that DNA methylation may affect assimilate loading in cucumber leaves (Figure 3E).

Comparative analysis results of CHH methylation site information and RT-qPCR data showed the complex relationship between DNA methylation and gene expression in the FNLs vs. NFNLs comparison. As shown in Figure 4A, in intergenic regions, the DNA methylation of *CsRS*, *BAK1* (*CsaV3_4G007620*) and *SAUR* (*CsaV3_6G007970*) was positively correlated with gene expression, whereas the methylation of *SPS* (*CsaV3_2G033300*), *IAA* (*CsaV3_2G013230*), *BRI1* (*CsaV3_5G015150*) and *BAK1* (*CsaV3_4G007620*) exhibited the negative regulation of gene expression. The gene body DNA methylation of *AUX1* (*CsaV3_5G002130*) was positively correlated with gene expression, and the DNA methylation of *SUS* (*CsaV3_4G000970*), *IAA* (*CsaV3_2G013230*) and *SAUR* (*CsaV3_6G007970*) showed a negative correlation with gene expression. In promoter regions, the upregulated *ERF1* (*CsaV3_3G012170*) was hypermethylated in FNLs, and the downregulated *SUS* (*CsaV3_4G000970*) had one hypermethylated region and one hypomethylated region. Moreover, the differentially expressed genes *AUX1* (*CsaV3_5G002130*) and *IAA* (*CsaV3_2G013230*) were positively and negatively correlated with DNA methylation in the exon regions, respectively. The DMRs and vital genes between FNLs and NFNLs are presented in the Integrative Genomics Viewer (IGV, Appendix A), including *SPS* (*CsaV3_2G033300*), *SUS* (*CsaV3_4G000970*), *ERF1* (*CsaV3_3G012170*), *CsGolS1* (*CsaV3_6G000050*), *CsRS* (*CsaV3_3G042690*) and *CsSTS* (*CsaV3_7G030090*), and the DMRs of six genes can be clearly retrieved in the genome. We also found that the same DEG may correspond to more than one DMR, and the methylation of the same DEG may occur in the intergenic region, upstream, in exons, etc., at the same time (Figure 4A, Appendix A). Overall, these results suggest that DEGs between FNLs and NFNLs were modified by DNA methylation in different regions, which might be a necessary condition for sink-source regulation.

We performed DNA methylation-sensitive restriction endonuclease digestion followed by PCR (Chop-PCR) to further investigate the differences in DNA methylation levels of the genes, including *CsRS* (*CsaV3_3G042690*), *AUX1* (*CsaV3_5G002130*), *SAUR* (*CsaV3_6G007970*), *ERF1* (*CsaV3_3G012170*), *BRI1* (*CsaV3_5G015150*) and *SUS* (*CsaV3_4G000970*), in NFNLs and FNLs (no available DNA methylation-sensitive restriction endonuclease sites were found in the DMR of other main genes) (Figure 4B). Methylated cytarabine protects the site from cleavage and allows for successful sequence amplification, whereas unmethylated DNA is fragmented and cannot perform PCR. Chop-PCR showed that *ERF1* (*CsaV3_3G012170*) and *CsRS* (*CsaV3_3G042690*) showed increased DNA methylation at the intergenic region in FNLs. *AUX1* (*CsaV3_5G002130*), *SAUR* (*CsaV3_6G007970*), *BRI1* (*CsaV3_5G015150*) and *SUS* (*CsaV3_4G000970*) displayed decreased methylation at the exon, intergenic, intergenic and upstream 2 kb regions, respectively, which further validated the accuracy of the WGBS results.

### 2.4. The Effect of 5-Aza-dC on Sink-Source Regulation

Moreover, 5-Aza-dC treatment on FNLs was performed to further investigate the effect of DNA methylation on cucumber sink-source regulation (Figure 5A). Changes in the expression level of the DNA methyltransferase gene or DNA demethylase gene may directly regulate the DNA methylation level. Based on the homology with the cytosine-5 DNA methyltransferase (C5-MTase) and DNA demethylase (dMTase) proteins in *Arabidopsis*, five C5-Mtase proteins and four dMTase proteins were identified in the cucumber genome (Figure 5B,C). The expression levels of these methylation-related genes in FNLs were examined under both CK and 5-Aza-dC treatment by RT-qPCR. Our results showed that, compared with the CK group, the gene expression of DNA methyltransferase (*CsMET)* domains, rearranged methylase 2a (*CsDRM2a*), and chromomethylase 2 (*CsCMT2*) were significantly downregulated in the 5-Aza-dC treatment group (Figure 5D). Additionally, the demethylase genes, a repressor of silencing (*CsROS*) and Demeter (*CsDME*) were upregulated and downregulated after 5-Aza-dC treatment, respectively (Figure 5D). These findings demonstrated that 5-Aza-dC treatment in cucumber plants could dynamically alter the DNA methylome pattern.

After 5-Aza-dC treatment on FNLs, the DNA methylation level of *CsRS* (*CsaV3_3G042690*), *AUX1* (*CsaV3_5G002130*), *SAUR* (*CsaV3_6G007970*), *ERF1* (*CsaV3_3G012170*), *BRI1* (*CsaV3_5G015150*) and *SUS* (*CsaV3_4G000970*) was depressed (Figure 6A). Furthermore, in FNLs, the relative expression level of *CsSTS* (*CsaV3_7G030090*) was highly upregulated in response to 5-Aza-dC treatment as well as CsSTS enzyme activity, while *CsGolS1* (*CsaV3_6G000050*) and *CsRS* (*CsaV3_3G042690*) were not significantly changed (Figure 6B,C). In addition, the content of stachyose in leaves was significantly upregulated (Figure 6D). After anthesis, the fruit length, fruit expansion rate and dry weight increased after 5-Aza-dC treatment for 1–7 days (Figure 6E,F), implying that the application of 5-Aza-dC has the potential to deregulate the repression of genes by DNA methylation and promote the loading of assimilates in cucumber leaves, thus increasing the fruit size.

## 3. Discussion

Coordinating the sink-source relationship is recognized as an essential way to increase crop production potential [32]. Together, hormones, sugars and environmental factors form an integrated signal network to balance the allocation between carbon source production and carbon sink utilization [5,33]. Indoleacetic acid (IAA) is an important chemical messenger to promote plant growth, and its large accumulation in source organs is conducive to regulating carbon translocation from leaves to grains [34]. Higher sink strength leads to complex changes in the expression of AUX1, IAA, GH3 and SAUR genes related to the auxin signal pathway in cucumber leaves, and their influence on sink-source balancing deserves further study. It was found that BRI1 anchored at the plasma membrane is responsible for initiating the BR signal. When BR is stimulated, BRI1 and BAK1 rapidly form the plasma membrane-anchored co-receptor complex to activate downstream signaling components [35]. Graeff et al. [36] reported that the *bri1 brl1 brl3* triple-mutant (lacking the three receptor kinases BRI1, BRI1-LIKE 1 (BRL1) and BRL3) in *Arabidopsis* can no longer sense brassinosteroid phytohormones and showed severe developmental defects in the phloem. Therefore, four upregulated *BRI1s* and three upregulated *BAK1* genes in FNLs might play a role in transporting signals between source and sink organs. Generally, higher levels of ethylene are a sign of leaf maturity [37]. The gene expression of most vital genes involved in ethylene biosynthesis and metabolism pathways increased in FNLs, indicating that higher sink activity may result in strengthened ethylene storage in cucumber leaves.

Sucrose synthases are a group of key enzymes responsible for catalyzing reversible reactions of sucrose decomposition and synthesis [38]. It has been reported that in the sucrose metabolism pathway, gene expression and enzyme activity associated with sucrose catabolism (SUS, invertase, etc.) decrease under low source-sink ratio conditions [16]. In our work, two *SUS* genes (*CsaV3_4G000970* and *CsaV3_5G020420*) were downregulated in FNLs, suggesting that a feedback regulatory mechanism in the sink-source relationship may lead to less sucrose being synthesized in leaves. In source leaves, trehalose 6-phosphate (T6P), which has been shown to inhibit sucrose-nonfermenting 1 (SNF1)-related kinase 1 (SnRK1) in vitro, regulates sucrose production to balance the supply and demand of sucrose from growing sink organs [39,40]. By regulating SnRK1, T6P stimulates the growth and division of meristem cells in response to high sucrose concentration stress and finally promotes starch accumulation [41]. In this study, two *TPP* genes were upregulated in FNLs, which might have downregulated T6P and then increased assimilate production in source organs. In addition, previous work shows that high sink strength leads to an increase in the stachyose production of cucumber source leaves [16], and the current study also confirms this finding.

It was generally accepted that DNA methylation could decrease the expression of genes and maintain genomic stability, and DNA demethylation increased levels of expression [42]. However, recent studies have shown that, in different genomic regions and different species, DNA methylation has different effects on the control of gene expression. For example, gene body methylation in rice and melon and gene promoter methylation in chickpeas are usually positively correlated with gene expression [43,44,45,46]. In watermelon and melon genomes, DNA methylation in the promoter region is usually negatively correlated with gene expression [45,47]. Nevertheless, DNA hypermethylation in wheat and apples, located in promoter and genomic regions, is not always in association with the suppression of gene expression [48,49,50]. The regulatory relationship between DNA methylation and gene transcription may be influenced by where DNA methylation occurs, the type of methylation, and other influences (such as transcription factors). So far, there is a lack of general evidence about whether there is a clear positive or negative correlation between DNA methylation and gene expression levels [51]. In the present study, in cucumber leaves under different sink strengths, DNA methylation in cds regions was negatively correlated with gene expression. In addition, a total of 77.8%, 60% and 53.1% of DMGs, which are methylated in exons, gene bodies and downstream 2 kb, are negatively correlated with gene expression. DNA methylation in 52.2% of genes, which were enriched in the upstream 2 kb, could promote gene expression (Appendix A). Therefore, in different species, DNA methylation in different regions may have different effects on gene expression. Additionally, Song et al. [52] speculated that the impact of DNA methylation on gene expression could be mediated by trans affects applied by some regulatory proteins, such as transcription factors. Therefore, the potential molecular mechanism of DNA methylation in regulating gene expression still needs to be further explored.

DNA methylation inhibitors are a class of chemicals that can reduce the level of genomic methylation and inhibit the occurrence of DNA methylation. Moreover, 5-Aza-dC is a commonly used inhibitor of DNA methylation enzymes in plant physiological research; it is able to covalently bind to DNA methyltransferases and reduce their catalytic activity, thus reducing the level of DNA methylation [48,53]. However, high concentrations of methylation inhibitors can lead to the yellowing and even death of plant organs [54,55,56]. Previous studies have shown that DNA methylation inhibitors in a certain concentration range can change DNA methylation levels and promote gene expression after treating rice, *Arabidopsis*, *Salvia miltiorrhiza* and strawberry [57,58,59,60]. In this study, four concentrations of 5-Aza-dC, 5 μM, 10 μM, 20 μM and 30 μM, were tested in a pretest study according to previous studies [61,62], and 20 μM of 5-Aza-dC was selected for treatment. While most of the previous studies were conducted by applying 5-Aza-dC in the culture medium to study the effect of DNA methylation on the growth and development of plant tissues and organs [48,60], in the present study, we sprayed the leaves directly with 5-Aza-dC and found the optimal concentration to promote the development of cucumber fruits. One of the reasons that 5-Aza-dC promotes fruit enlargement may be that 5-Aza-dC treatment alters the gene expression of C5-MTase and dMTase, thereby altering the DNA methylation level of genes associated with sink-source carbon partitioning to facilitate assimilate loading. In addition, transcription factors could combine with specific sites of targets and regulate the expression of downstream target proteins [63]. DNA methylation is reported to alter the expression of transcription factors, thereby enabling the transcriptional regulation of target genes [61,62]. In our study, the differentially expressed transcription factors between FNLs and NFNLs, including MYBs, WRKYs, HSFs (heat shock transcription factors) and DREBs (dehydration-responsive element-binding proteins), may bind to upstream promoters of *CsGolS1*, *CsRS* and *CsSTS* (Appendix A). Therefore, DNA methylation and demethylation, which occurred in these transcription factors after 5-Aza-dC treatment, may activate the gene expression of assimilate loading in FNLs and promote fruit swelling.

DNA methylation can change dynamically during plant growth and development and under the stress of different potential environmental factors [64,65]. For example, heat stress leads to massive deletion of genome-wide DNA methylation in woodland strawberry [25]. Extreme water deprivation leads to DNA hypomethylation and gene up-regulation in the resurrection plant *Boea hygrometrica* [66]. Genomic methylation variation is associated with leaf shape and photosynthetic properties in natural populations of *Populus simonii* [67]. Whereas the present study found that under conditions of varying sink strengths (removing all fruits or retaining one fruit on a plant), it was also able to alter the DNA methylation pattern in cucumber source leaves. Moreover, DNA methylation has been studied in more detail in sucrose-transporting plants [30,31], but very little in RFO-transporting plants. Meanwhile, we externally applied 5-Aza-dC, a widespread DNA methylation inhibitor, on FNLs, and found that it could increase stachyose synthase (*CsSTS*) gene expression, enzyme activity and stachyose content in the RFO synthesis pathway of source leaves and increase fruit length and dry weight, which further verified that DNA methylation promotes fruit enlargement by regulating RFO synthesis. This finding provides an innovative reference for the study of sink-source relationships in DNA methylation-regulated RFO-transporting crops (e.g., melon, watermelon, pumpkin, and winter melon). However, we used only one representative East-Asian line of cucumber, “Jinchun 5,” to study the regulation of DNA methylation on the cucumber sink-source relationship. Therefore, we will add other cucumber core lines to our next research program in order to explore more specific and comprehensive regulatory mechanisms of DNA methylation in relation to the sink-source relationship.

## 4. Materials and Methods

### 4.1. Plant Materials and Treatments

In this study, Jinchun 5 cucumber seeds were obtained from the Tianjin Cucumber Institute (Tianjin, China). The germinated seeds were sowed in 50-well pot trays. At the three-leaf stage, the cucumber seedlings were planted in 40 cm × 40 cm plastic pots (Each pot contained 8 kg of air-dried soil) in growth chambers under 14 h light at 24 °C and 10 h dark at 16 °C, with a relative humidity of 70%. The experiment was performed at the experimental station of Yangzhou University, Jiangsu Province, China, in the spring of 2022. Yamazaki cucumber nutrient solution was fertilized weekly [68]. Different sink demands were achieved by removing all fruits or retaining one fruit on a plant. Leaves without or with a single fruit (NFNL or FNL) at the 12th node were collected at eight days post anthesis (dpa) and stored at −80 °C. A total of 25 NFNL- or 25 FNL-treated plants were divided into five biological replicates. Five NFNLs or FNLs were taken from different plants from the same plot and mixed as one sample.

### 4.2. WGBS, RNA Sequencing and Data Analysis

Three biological replicates of FNLs and NFNLs collected above were sent to E-gene (www.egenetech.com, Shenzhen, China, accessed on 9 March 2023) for WGBS. Three micrograms of genomic DNA was sonicated to 100–300 bp and treated with bisulfite to construct the libraries on an Illumina HiSeq™ 2500 platform. The sequence data had high conversion efficiency and complied with relevant standards (Appendix A). BSMAP (version 2.73) [69] was used to perform alignments of clean reads to the cucumber genome [70] using default parameters. DMRs were identified by Metilene v0.2-7 [71]. The cut-off of methylation analysis was set to *p* < 0.05 using the 2D KS test to detect significant DMRs. The genome sequence was divided into upstream 2 kb, the gene body, exon, cds, and downstream 2 kb according to different gene structural elements.

The sample processing of RNA sequencing was the same as that of WGBS. Total RNA was extracted from 0.1 g FNLs and 0.1 g NFNLs by RNAiso Plus (Takara, Beijing, China). High-quality RNAs of OD_260_/OD_280_ = 1.8–2.4 and OD_260_/OD_230_ = 1.5–2.4 were used for library construction. After deep sequencing on an Illumina HiSeq™ 2500 platform by Biomarker (Beijing, China), clean reads were aligned to the cucumber genome [70] using a sequence alignment software HISAT 2.2.1 [72]. Edge R was applied to analyze the differentially expressed (DE) mRNAs between FNLs and NFNLs, with a fold change of ≥2 and an adjusted *p* value of <0.05 as filtering criteria [73].

### 4.3. Chop-PCR

For Chop-PCR, genomic DNA was extracted from three biological replicates of FNLs and NFNLs using a Plant Genome DNA Kit DP305 (Tiangen, Beijing, China), and 1 μg genomic DNA was digested overnight with 1μL of a methylation-sensitive restriction endonuclease (ClaI, AvaI, DpnI, SnaBI, FspEI or HpaII (NEB, Ipswich, MA, USA)) in a 30 μL reaction mixture. Using 1 μL of the digested DNA, PCR was performed in a 10 μL reaction mixture with 5 μL of 2 × *EasyTaq*^®^ PCR SuperMix (+dye) (TransGen Biotech, Beijing, China), 0.5 μL of forward primer, 0.5 µL for the reverse primer and 3 μL double-distilled water (ddH_2_O). Undigested DNA was amplified as the loading control. The sequences amplified by the Chop-PCR primers contained the DMR and endonuclease sites, which are listed in Appendix A.

### 4.4. Treatment with DNA Methylation Inhibitor 5-Aza-dC

Twenty micromoles per liter DNA methylation inhibitor 5-Aza-dC (Sigma-Aldrich, a2385-1 g, Shanghai, China, dissolved in 30% (v/v) dimethyl sulfoxide, DMSO), was applied to the adaxial and abaxial leaf surfaces of FNLs at anthesis. FNL treated with 30% (v/v) DMSO was used as the control (CK). Cucumber fruit length and dry weight were recorded from 0–7 dpa. CK- and 5-Aza-dC-treated leaves were sampled before 8 dpa. Five independent biological replicates were performed for 5-Aza-dC and the control treatments.

### 4.5. Sugar Content and Enzyme Activity Determination

Soluble sugars, including stachyose, raffinose, sucrose, glucose and fructose, were extracted from five biological replicates of FNLs and NFNLs. High-performance liquid chromatography (HPLC) was used to determine the sugar contents, which were expressed as mg per g fresh weight (FW; tissue) [74]. Briefly, a 0.3 g fresh tissue sample was ground with liquid nitrogen and then extracted three times each time with 5 mL 80% ethanol. The extracts were dried under vacuum with a Buchler Evapo-Mix (Buchler Instruments, Fort Lee, NJ, USA). The contents were dissolved in 1 mL ddH_2_O, centrifuged, and analyzed by HPLC. The five sugar contents were determined by comparing the retention times of sugar standards.

The enzyme activity of CsGolS, CsRS and CsSTS from five biological replicates of FNLs and NFNLs were assayed according to our previous works [17]. Enzyme activity is expressed as μmol of CsGolS, CsRS or CsSTS formation per hour per gram of protein. Bovine serum albumin was applied as a standard to determine the protein concentration of the FNLs and NFNLs.

### 4.6. Genome-Wide Identification and Phylogenetic Analysis of the C5-MTase and dMTase Genes in Cucumber

To identify the C5-MTase and dMTase genes in the cucumber genome, the amino acid sequences of C5-MTase and dMTase in *Arabidopsis* were downloaded from TAIR10 and used as queries in BLASTp (*E* value < 1 × 10^−5^) searches against the cucumber genome [70] (Appendix A). A CDD search [75] was used to confirm the accuracy of these amino acid sequences of cucumber C5-MTase and dMTase, and sequences without conserved domains were excluded. The protein sequences of C5-MTase and dMTase in *A. thaliana* and *C. sativus* were aligned using MEGA11 [76], and 1000 bootstrap replicates were set to generate a neighbor-joining (NJ) tree. Subsequently, iTOL [77] was applied to visualize the phylogenetic tree.

### 4.7. RNA Extraction and RT-qPCR

Total RNA was isolated from NFNLs, FNLs, the 5-Aza-dC treatment of FNL and CK, as described in Section 2.2. Primers for RT-qPCR and the internal control were designed using Primer Premier 5. RT-qPCR primers, the stably expressed reference gene (18S rRNA, GenBank accession No.: AF206894) and cycling conditions are listed in Appendix A. The reaction was performed on a CFX Connect Real-Time System (Bio-Rad, Hercules, CA, USA). The 2^−ΔΔCT^ method was used to calculate relative gene expression levels, and three biological replicates with three technical replicates were set [78].

### 4.8. Statistical Analysis

The data are presented as the mean values ± standard deviation (SD) of three biological replicates in an RT-qPCR experiment and of five biological replicates in the determination of enzyme activity, soluble sugar content, fruit length and dry weight. The Student’s *t*-test was applied for the statistical difference comparison using GraphPad Prism 9 (GraphPad Software, San Diego, CA, USA). Differences of *p* < 0.05 (*), *p* < 0.01 (**) and *p* < 0.001 (***) were significant.

## 5. Conclusions

Photosynthate partitioning between sources and sinks is a vital determinant of cucumber yield. This study provides the first insights into the epigenetic landscape under different sink strengths in cucumber. Methylome and transcriptome analysis results reveal that DEGs involved in auxin, ethylene and brassinolide biosynthesis and metabolism; sucrose metabolism; and RFO synthesis pathways were modulated by DNA methylation. Furthermore, the application of 5-Aza-dC in FNLs promoted fruit growth. Also, 5-Aza-dC upregulated the relative expression of *CsSTS*, the enzyme activity of CsSTS, and the stachyose content involved in leaf assimilate loading, suggesting an essential role of 5-Aza-dC in sink-source regulation. This study reveals the contribution of DNA methylation to sink-source relationships and raises the possibility of improving cucumber yield by altering DNA methylation.

## Figures and Tables

**Figure 1 plants-13-00103-f001:**
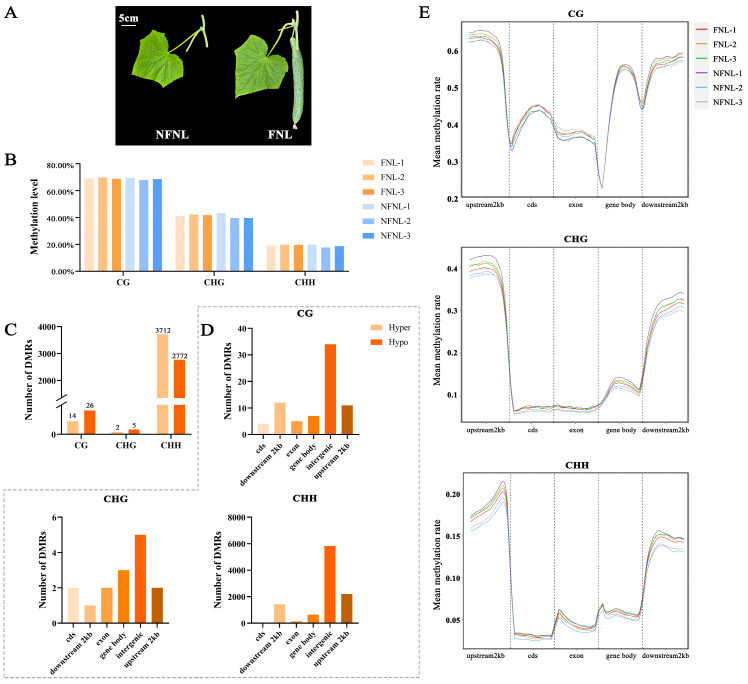
The genome-wide DNA methylation pattern and number of differentially methylated regions (DMRs) in FNLs and NFNLs. (**A**) Cucumber leaves phenotypes of FNLs and NFNLs. (**B**) Average DNA methylation level in CG, CHG, and CHH contexts. (**C**) Characteristics of the methylation levels of gene elements in CG/CHG/CHH contexts. (**D**) Number of differentially methylated regions (DMRs) in CG, CHG, and CHH contexts. (**E**) The number of DMRs in cds, downstream 2 kb, exon, the gene body, intergenic and upstream 2 kb in CG/CHG/CHH contexts.

**Figure 2 plants-13-00103-f002:**
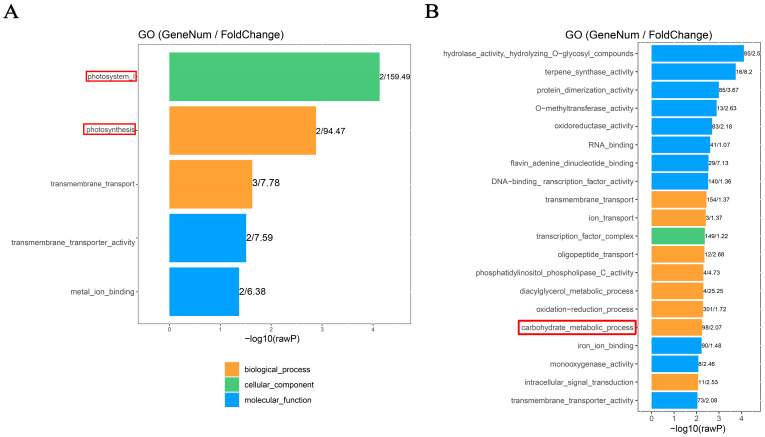
GO enrichment analysis of differentially methylated genes (DMGs) between FNLs and NFNLs in cucumber. (**A**) GO enrichment of differentially methylated genes (DMGs) in the CG context. (**B**) GO enrichment of DMGs in the CHH context.

**Figure 3 plants-13-00103-f003:**
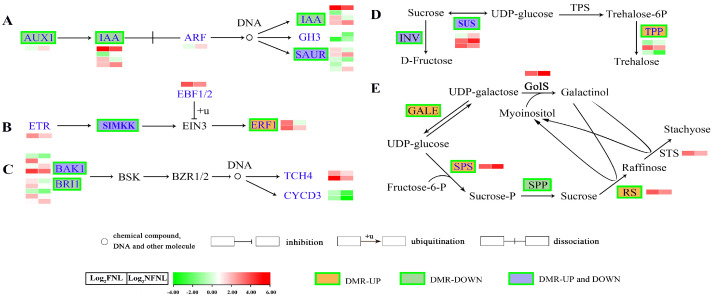
Comparative analysis of differentially methylated regions and differentially expressed genes in the metabolism pathways related to sink-source regulation. (**A**) auxin; (**B**) ethylene; (**C**) brassinolide (BR) biosynthesis and metabolism pathways. AUX1, auxin influx carrier (AUX1 LAX family); IAA, auxin−responsive protein IAA; ARF, auxin response factor; SAUR, SAUR family protein; GH3, auxin−responsive GH3 gene family; ERF1, ethylene−responsive transcription factor 1; ETR, ethylene receptor [EC:2.7.13.−]; EBF1/2, EIN3−binding F−box protein; SIMKK, mitogen−activated protein kinase 4/5 [EC:2.7.12.2]; BRI1, protein brassinosteroid insensitive 1 [EC:2.7.10.1 2.7.11.1]; BSK, BR-signalling kinase [EC:2.7.11.1]; BAK1, brassinosteroid insensitive 1−associated receptor kinase 1 [EC:2.7.10.1, 2.7.11.1]; BZR1/2, brassinosteroid resistant 1/2; CYCD3, cyclin D3, plant; TCH4, xyloglucan: xyloglucosyl transferase TCH4 [EC:2.4.1.207]. (**D**) Sucrose metabolism pathway. SUS, sucrose synthase [EC:2.4.1.13]; INV, beta−fructofuranosidase [EC:3.2.1.26]; TPS, trehalose 6−phosphate synthase [EC:2.4.1.15; 2.4.1.347]; TPP, trehalose 6−phosphate phosphatase [EC:3.1.3.12]. (**E**) RFO synthesis pathway. GALE, UDP−glucose 4−epimerase [EC:5.1.3.2]; SPS, sucrose−phosphate synthase [EC:2.4.1.14]; SPP, sucrose phosphate phosphatase [EC:3.1.3.24]; GolS, galactinol synthase [EC:2.4.1.123]; RS, Raffinose synthase [EC:2.4.1.82]; STS, stachyose synthetase [EC:2.4.1.67]. Green boxes denote that the gene is modified by a DMR. The heatmap shows the expression levels of differentially expressed genes (DEGs) in NFNLs and FNLs in RNA−seq. Log_2_ (FNL/NFNL) represents the gene fold-change, where FNL/NFNL refers to the ratio of FPKM reads.

**Figure 4 plants-13-00103-f004:**
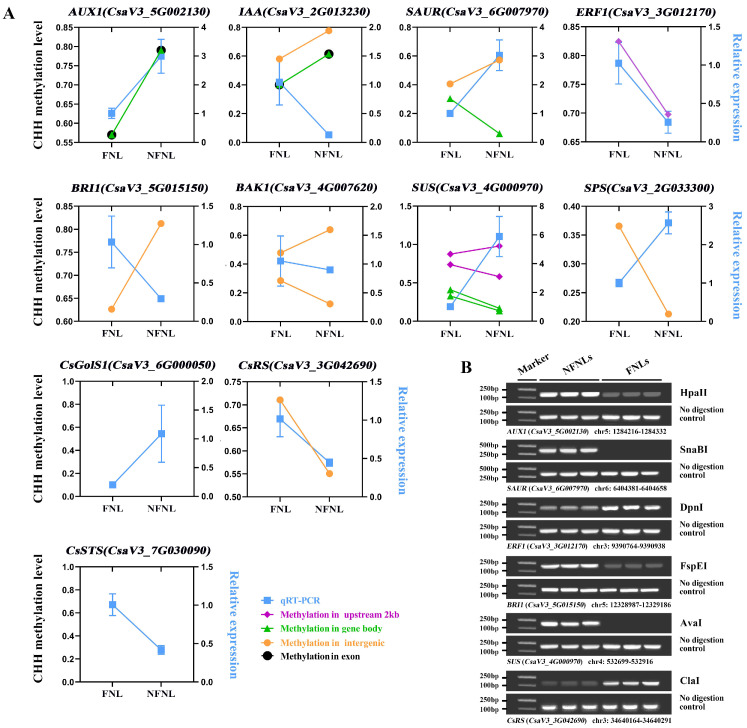
Correlation between DNA methylation enrichment and the expression level of genes related to sink-source regulation between NFNLs and FNLs. (**A**) The ordinates on the left and right indicate the DNA methylation level in the CHH context and the relative expression level between NFNLs and FNLs, respectively. Data are presented as the means ± SDs (*n* = 3) in RT-qPCR. (**B**) DNA methylation-sensitive restriction endonuclease digestion followed by PCR (Chop-PCR) validation of DNA methylation levels between NFNLs and FNLs. ClaI, AvaI, DpnI, SnaBI, FspEI and HpaII are methylation-sensitive restriction endonucleases. Amplification of non-digested DNA served as a control.

**Figure 5 plants-13-00103-f005:**
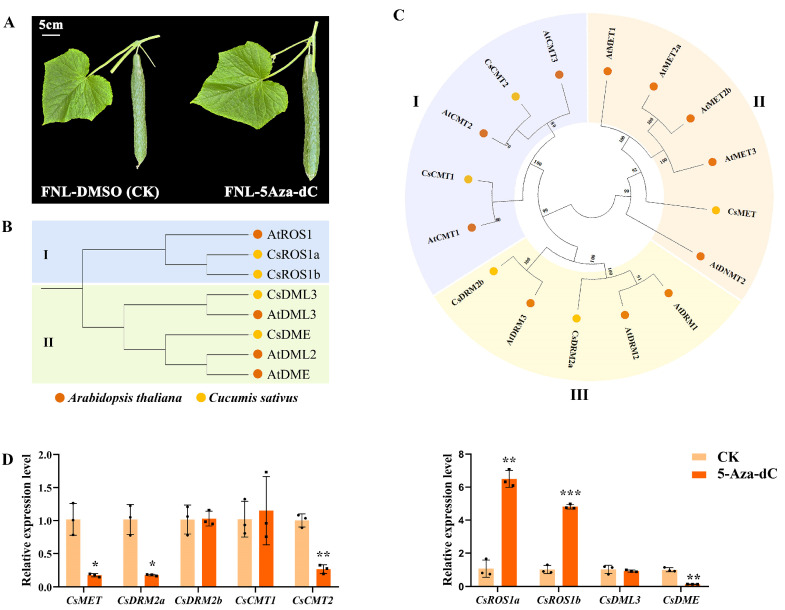
Expression profiling of the cytosine-5 DNA methyltransferase (C5-MTase) and DNA demethylase (dMTase) in FNLs after 5-aza-dC-2′-deoxycytidine (5-Aza-dC) treatment. (**A**) Fruit phenotypic characterization of 5-Aza-dC treatment and control groups on FNLs. (**B**) Phylogenetic tree representing the relationship between dMTase genes of *A. thaliana* and *C. sativus*. Circles of different color represent different species. I and II represent the two subfamilies. (**C**) Phylogenetic tree representing the relationship between C5-MTase genes of *A. thaliana* and *C. sativus*. I–III represent the three subfamilies. (**D**) Expression analysis of methyltransferase genes and demethylase genes in FNLs treated with and without 5-Aza-dC. Data are presented as the means ± SDs (*n* = 3) in RT-qPCR. The asterisks represent significant differences relative to the CK group (* *p* < 0.05; ** *p* < 0.01; *** *p* < 0.001 using Student’s *t*-test).

**Figure 6 plants-13-00103-f006:**
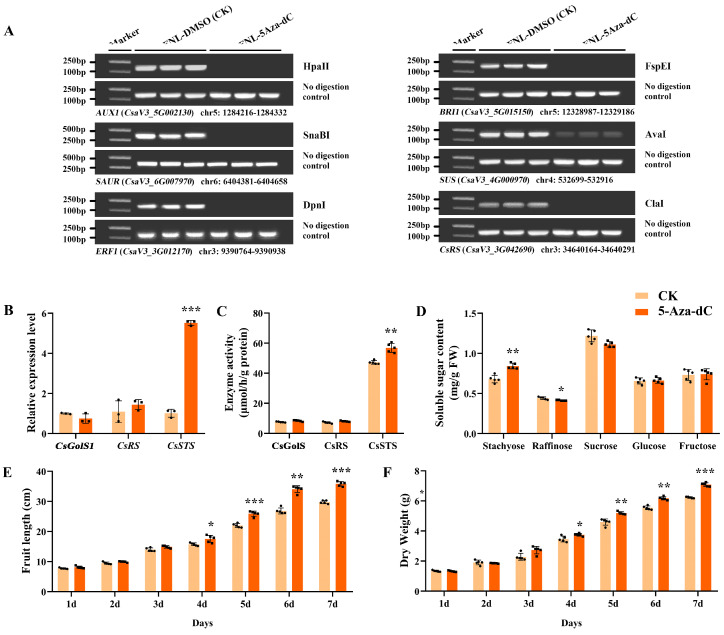
Expression profiles and DNA methylation levels of sink-source related genes on leaves and fruit phenotypic characterization analysis after 5-Aza-dC treatment on FNLs. (**A**) Analysis of DNA methylation status between 5-Aza-dC treatment on FNLs and control groups by Chop-PCR. (**B**) The relative expression levels of *CsGolS1*, *CsRS* and *CsSTS* in FNLs. (**C**) Enzyme activity of CsGolS, CsRS and CsSTS in FNLs. (**D**) Soluble sugar content of FNLs. (**E**) Fruit length. (**F**) Dry weight of fruit. Data are expressed as the means ± standard deviation (SD) of three replicates in figures (**B**–**D**) and of five replicates in figures (**E**,**F**). Error bars represent the standard deviation among three/five independent replicates. The asterisks represent significant differences relative to the CK group (* *p* < 0.05; ** *p* < 0.01; *** *p* < 0.001 using Student’s *t*-test).

## Data Availability

Data of WGBS and RNA sequencing can be found in NCBI (accession: PRJNA938623).

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
