# Peer review of "The Sink-Source Relationship in Cucumber (*Cucumis sativus* L.) Is Modulated by DNA Methylation"

_plants, 2023, doi:10.3390/plants13010103_

Round 1
Reviewer 1 Report
Comments and Suggestions for Authors
The manuscript titled “The sink‒Source Relationship in Cucumber (Cucumis sativus L.) is Modulated by DNA methylation” reports an interesting work discussing the the epigenetic genomic landscape of cucumber leaves with different sink strengths by whole-genome bisulfite sequencing. Authors reported the relationship between the enrichment level of DNA methylation and the transcription level of genes in auxin, ethylene and brassinolide biosynthesis and metabolism, and sucrose and RFO synthesis metabolism involved in the sink‒source relationship. Application of DNA methylation inhibitor 5-aza-dC-2'-deoxycytidine (5-Aza-dC) on the leaves carrying the fruit revealed that 5-Aza-dC can depressed the DNA methylation levels of sink-source related genes, increase the expression, activity and content of CsSTS. This is a well-written article and I anticipate that the manuscript should be of great interest to the researchers working on epigenetics and DNA methylation. I include my comments below, most of them are suggestion to improve the overall quality for publication. I considered the manuscript suitable for publication subject to following improvements.
Specific Comments
Overall, the study is well designed and presented in a good way. However, many sentences include repetitive words and not explained and cited appropriately.
The authors elaborated abstract in a good way. However, some prominent results should be added to improve this section.
Rearrange the keywords alphabetically.
Introduction section should be more robust by adding recent references.
The objectives of the study should be revised and readers friendly.
Does the study explore the long-term effects of altered DNA methylation on the health and viability of the plants?
Can you add some information regarding potential environmental factors that could influence DNA methylation patterns and, consequently, sink‒source regulation in different growing conditions?
Is there any potential unintended consequences of manipulating DNA methylation for the purpose of increasing crop yield?
How generalizable are these findings to other crops?
Line 72-77: restructure the statements, please
Line 81-84: Connect the two paras, require a statement which link the paragraphs.
Literature cited and discussed in a proper way (well written and presented).
Incorporate some latest references.
The article should become acceptable after minor revisions of English / grammatical mistakes.
Reviewer 2 Report
Comments and Suggestions for Authors
Here are my comments:
- Expand the background information on the significance of cucumber as a crop for studying sink-source relationships, highlighting what sets it apart from other model plants.
- Clarify the novel aspects of this study compared to existing research in plant epigenetics, particularly in the context of sink-source dynamics in cucumbers.
- Provide more detailed analysis and interpretation of figures that are under-discussed in the manuscript, such as Figure S1 and S2.
- Rebalance the manuscript by expanding the discussion section to better elaborate on the implications and significance of the findings.
- Offer more explicit details about the methodologies used, including the rationale behind specific experimental conditions and the choice of concentrations for 5-Aza-dC.
- Discuss potential limitations or biases in the experimental design and methodology and how these might influence the results and interpretations.
- Improve the manuscript's language and syntax to enhance clarity and readability, correcting grammatical and syntactical errors.
- Ensure consistent use of terminology and definitions throughout the manuscript to maintain clarity and avoid confusion.
- Provide a comparative analysis with previous studies to emphasize the unique contributions and advancements made by this research.
- Discuss the broader implications of the findings for plant biology and crop science, highlighting the potential impact on future research.
- Strengthen the synthesis of results with existing knowledge in the discussion section, underscoring the study's relevance and potential influence on future research directions.
- Given the identified gaps, recommend a major revision to enhance the manuscript's quality and readiness for academic publication.
Round 2
Reviewer 1 Report
Comments and Suggestions for Authors
Accept in present form